# Surface Wettability and Electrical Resistance Analysis of Droplets on Indium-Tin-Oxide Glass Fabricated Using an Ultraviolet Laser System

**DOI:** 10.3390/mi12010044

**Published:** 2021-01-01

**Authors:** Hsin-Yi Tsai, Chih-Ning Hsu, Cheng-Ru Li, Yu-Hsuan Lin, Wen-Tse Hsiao, Kuo-Cheng Huang, J. Andrew Yeh

**Affiliations:** 1Taiwan Instrument Research Institute, National Applied Research Laboratories, Hsinchu 300092, Taiwan; kellytsai@narlabs.org.tw (H.-Y.T.); bojik627@narlabs.org.tw (C.-N.H.); chengru@narlabs.org.tw (C.-R.L.); marklin@narlabs.org.tw (Y.-H.L.); wentse@itrc.narl.org.tw (W.-T.H.); 2Department of Power Mechanical Engineering, National Tsing Hua University, Hsinchu 300092, Taiwan; jayeh@pme.nthu.edu.tw

**Keywords:** indium tin oxide (ITO), wettability, electric resistance, hydrophilic, 355-nm UV laser, surface treatment

## Abstract

Indium tin oxide (ITO) is widely used as a substrate for fabricating chips because of its optical transparency, favorable chemical stability, and high electrical conductivity. However, the wettability of ITO surface is neutral (the contact angle was approximately 90°) or hydrophilic. For reagent transporting and manipulation in biochip application, the surface wettability of ITO-based chips was modified to the hydrophobic or nearly hydrophobic surface to enable their use with droplets. Due to the above demand, this study used a 355-nm ultraviolet laser to fabricate a comb microstructure on ITO glass to modify the surface wettability characteristics. All of the fabrication patterns with various line width and pitch, depth, and surface roughness were employed. Subsequently, the contact angle (CA) of droplets on the ITO glass was analyzed to examine wettability and electrical performance by using the different voltages applied to the electrode. The proposed approach can succeed in the fabrication of a biochip with suitable comb-microstructure by using the optimal operating voltage and time functions for the catch droplets on ITO glass for precision medicine application. The experiment results indicated that the CA of droplets under a volume of 20 μL on flat ITO substrate was approximately 92° ± 2°; furthermore, due to its lowest surface roughness, the pattern line width and pitch of 110 μm exhibited a smaller CA variation and more favorable spherical droplet morphology, with a side and front view CA of 83° ± 1° and 78.5° ± 2.5°, respectively, while a laser scanning speed of 750 mm/s was employed. Other line width and pitch, as well as scanning speed parameters, increased the surface roughness and resulted in the surface becoming hydrophilic. In addition, to prevent droplet morphology collapse, the droplet’s electric operation voltage and driving time did not exceed 5 V and 20 s, respectively. With this method, the surface modification process can be employed to control the droplet’s CA by adjusting the line width and pitch and the laser scanning speed, especially in the neutral or nearly hydrophobic surface for droplet transporting. This enables the production of a microfluidic chip with a surface that is both light transmittance and has favorable electrical conductivity. In addition, the shape of the microfluidic chip can be directly designed and fabricated using a laser direct writing system on ITO glass, obviating the use of a mask and complicated production processes in biosensing and biomanipulation applications.

## 1. Introduction

Microfluidics technology refers to the precision control and manipulation of fluids at the submillimeter scale. Generally, microfluidic systems transport, separate, and mix the fluids, and this technology has been applied in deoxyribonucleic acid (DNA) chip, micropropulsion, microthermal, and lab-on-a-chip technology [1,2,3]. Generally, soft lithography technology [4] is used to define the pattern and channel on substrates, such as wafer or glass. However, the pattern must be designed and the mask must be manufactured using the soft lithography and wet etching process. The mask’s flexibility is relatively low because it must be remade when the shape of the designed pattern changes. Hence, a dry patterned and maskless method, such as laser direct writing, is employed to determine the pattern contour; such technology is crucial for defining the pattern and fabricating microstructures on substrates as part of the development of microfluidic chips. Therein, the femtosecond laser is used to directly fabricate the three-dimensional (3D) microstructure in the glass [5] or polymeric microfluidic chip [6]. The microstructures comprise the microchannel, microsieves, microvalves, and micromixers [7]. Related studies have revealed that the femtosecond laser is a powerful 3D processing tool with the following advantages: high spatial resolution, high accuracy, and high compatibility with other processing techniques. After the pattern shape is determined on the substrate, glass and polydimethylsiloxane (PDMS) are widely used as the main material because they are transparent and have been widely applied in the biochip or microfluidic systems in recent years [8]. The advantages of glass-based and PDMS based chips are their high transparency, favorable biocompatibility, and chemical stability. Zhang et al. [9] used dry film photoresists as the pattern transfer masks for wet etching to fabricate glass-based microfluidic devices. This method requires less expensive materials and facilities compared with conventional methods, such as producing a glass mask by scanning electron microscopy or focused ion beam technology. In addition, the glass molding process was used to fabricate microstructures, and the roughness and height of the glass material were analyzed to determine the chip performance in one study [10]. The results indicated that the microstructure of the glass microfluidic chip could be fabricated with high surface quality; thus, the method can be applied to the mass production of glass fluidic chips. García et al. [11] presented a laser-based process to fabricate the microstructures on glass for reshaping and improving the morphological and optical qualities for microfluidic applications. Therein, the manufactured chips were used to capture the circulating endometrial tumor cells lines with Hec 1A tested tumor cell after coating the anti-epithelial cell adhesion molecule (EpCAM)antibody. The cells were successfully arrested on the pillars with the laser processing method, and it can give the technology a translational application in the field of cancer research. In addition to conventional glass material, indium tin oxide (ITO)—a transparent and colorless material in thin film—has been widely applied in microfluidic devices because of its optical transparency and electrical conductivity characteristics. The electrical conductivity of ITO films enables the widespread production of electrophoresis or dielectrophoresis biochips in polymerase chain reaction (PCR) or particle/cell screening applications [12,13]. Ghrera et al. [14] used the chemical etching method to produce a pattern on ITO-coated glass and seal it with PDMS to fabricate the microchannel. In addition, the ITO substrate was modified by depositing cadmium selenide quantum dots (QCdSe) by using the Langmuir–Blodgett technique, and the QCdSe surface was functionalized with a specific DNA probe. The interfacial charge transfer resistance value was measured in the hybridization process to monitor the concentration of target complementary DNA. Konry et al. [15] fabricated a micropatterned structure of conductive polymer on an ITO glass chip for the subsequent photoimmobilization of various bioreceptors and antigens. The results revealed the resolution of electropolymerization on an ITO pattern and that more than one bioreceptor can be simultaneously detected in several analytes. The microarray chip exhibited favorable sensitivity and selectivity, comparable to that of standard hospital detection methods. Huang et al. [16] used PDMS and ITO glass with electrodes to fabricate dielectrophoretic microfluidic devices and applied a microfluidic chip and a positive dielectrophoretic force to capture and screen sperm for manipulating the oocyte. With the developed microfluidic device, the oviduct of mammals could be imitated with the microchannel to achieve the fertilization in vitro of an imprinting control-region mouse. The aforementioned studies indicate that ITO glass has been widely applied in microfluidic chips as the electrode to drive the droplet and fluids, including cell reagents or bioreagents. Hence, the wettability of the surface of ITO glass is a critical topic, that is, whether the hydrophobic surface is suitable for a droplet-type chip and whether the hydrophilic surface can be applied in a microchannel chip.

Wettability, which is a crucial property to determine the contact ability of a liquid on a solid surface, results from the intermolecular interactions of the intrinsic surface energy and surface morphology when the liquid and solid surface are brought together [17,18]. Numerous studies have focused on the control of wettability of real surfaces, which can be divided into hydrophilicity and hydrophobicity according to the contact angle (CA) [19]. Surfaces that are superhydrophobic or superhydrophilic may act as self-cleaning surfaces or be employed as antibiofouling surfaces in biomedical applications, respectively [20,21]. Therefore, the wettability of substrate’s surface was important to the biomedical chip application. When the surface was a hydrophobic, the droplet was easily to be moved and could be manipulated to the specific region on the chip. Various methods can be employed to modify the neutral surface of a material into a hydrophobic or hydrophilic surface, for example, by employing a mechanical method, chemical treatment, or thin film coating. The surface roughness can be controlled and the wettability can be alternated by employing a well-designed microstructure on the surface [22,23]. The most common example is the lotus-leaf effect, in which the numerous microstructures present on the lotus leaf result in the surface being superhydrophobic. The CA of a droplet on the surface can be enhanced to >160°. Laser processing is another common method used in industry and research to change the surface roughness and wettability of material; in this type of processing, microstructures with different patterns and depths can be designed and directly interact on the material surface to obtain different levels of surface wettability [24]. Ryu et al. [25] used the laser-ablation-assisted patterning method on a graphene/ITO double layer to fabricate the electrode pattern, which induced a more effective thermal-energy transfer reaction during laser ablation compared with ITO single layers. With this method, transparent and high-performance electronic devices can be fabricated for transparent conductive electrodes and various other applications. Tsai et al. [26] used an ultraviolet (UV) laser with a 355-nm wavelength to fabricate a mesh pattern on a rigid gas permeable (RGP) contact lens (Boston XO) to modify its surface properties. The results revealed that the CA of the droplet on the lens material decreased approximately 10°–20° when the pitch of the mesh pattern was less than 50 μm under a laser scanning speed of 500 mm/s. In addition, the CA decreased to 40° with laser treat and without laser treat surface after oxygen plasma treatment. However, the CA of droplet on the surface treated with UV laser, and oxygen plasma was twice that of the surface treated only with oxygen plasma. Therefore, the edge of RGP contact lenses can be treated with UV laser and oxygen plasma to enhance their tear wettability and wearing comfort.

Several studies have focused on the wettability of ITO glass or the effect of laser treatment on surface modification; however, no related study has examined the wettability of ITO glass that has undergone laser surface modification and been subjected to an electric field. Laser technology offers advantages, such as rapidity and pattern flexibility, when used on ITO glass substrates; thus, the shape of the ablated pattern can be directly designed on software in future applications. Therefore, we employed a UV laser system to fabricate a comb pattern and hoped to obtain a hydrophobic surface on the ITO glass, which included two electrode pads that could be used to exert an electrical charge on the droplet to observe its wettability. The potential volume of the droplet was selected through unpatterned ITO glass surfaces to generate a droplet shape. In addition, the laser system’s processing parameters and microstructure morphology (e.g., scanning speed and ablated line pitch and depth) were adjusted, and their effect on the surface roughness, depth, and conductivity of the ITO glass, as well as the CA of the droplet, was analyzed. In this manner, we determined the optimal laser-ablation parameters for the fabrication of a neutral (contact angle was approximately 90°) or hydrophobic surface and the patterns that can withstand higher voltages for regent manipulation and biosensing application. Our experimental results revealed that laser ablation can be used to fabricate neutral or hydrophobic surfaces on ITO glass to maintain the droplet’s spherical morphology. This process can expand the applicability and application field of ITO glass in biochips for the precision medicine field.

## 2. Fundamental Theory

A hydrophobic surface enables a reagent to combine with a bioparticle or sample to form a droplet; the reagent can then be heated, cooled, and mixed by exerting a voltage on the droplet. The CA, which generally represents the wettability of a solid surface by a liquid, is defined by the angle between the liquid–vapor interface and solid–liquid interfaces [27,28]. In general, the shape of the liquid–vapor interface is determined by Young’s equation [29]. The theoretical description of CA considers a thermal dynamic equilibrium between three phases: solid, liquid, and vapor. The solid–vapor interfacial energy is defined by *γ*_SG_, and the interfacial energy between the solid–liquid and the liquid–vapor interface are defined as *γ*_SL_ and *γ*_LG_, respectively (Figure 1). Then, the equilibrium CA (or *θ_C_*) is determined according to the quantities in the Young equations and written as Equation (1):(1)$$\gamma_{\mathrm{SG}}-\gamma_{\mathrm{SL}}-\gamma_{\mathrm{LG}}\cos\theta_{C}=0$$

Generally, a CA greater than 90° indicates that the solid material has a poor wettable surface (i.e., a hydrophobic surface); if the CA is greater than 150°, the material has a superhydrophobic surface. In superhydrophobic materials, liquids cannot enter the solid surface of the microstructure, resulting in a small interface between the solid material and the liquid; this phenomenon is called the lotus-leaf effect. By contrast, when the CA is less than 90°, the material has a hydrophilic surface and favorable wettability. In addition, the surface was defined as neutral when the CA is equal 90°.

Microstructures provide surface roughness and can be divided into homogeneous and heterogeneous microstructures. If a solid material has a rough surface, the liquid has close contact with the microstructures on the solid surface, and the droplet would be in the Wenzel state [30,31,32]. However, the droplet would be in the Cassie–Baxter (CB) state if it rests on the tops of microstructures (Figure 2). When droplets are placed on the surface of microstructures, different wetting states (e.g., Wentzel state and CB state) may occur depending on the surface materials, dimension of the microstructures, size of the droplets, thermal fluctuations, and external stimuli (e.g., voltage or electric field). Zhang et al. [33] used molecular dynamic simulations to observe the mechanism behind the CB-to-Wenzel transition of a nanoscale water film placed on a single nanogroove surface under the effect of an external electric field. The simulation snapshots and experimental results revealed that the water infiltrated into the groove, which reduced the CA; this provides direct evidence of electric-field-induced CB-to-Wenzel transition. If the surface of the material treated with the laser system is maintained as a homogeneous surface and not subjected to an electric field, the phenomenon can be described by the Wenzel mode (Figure 2a), as written in Equation (2):(2)$$\cos\left(\theta^{*}\right)=r\;\cos\left(\theta \right),$$
where *θ** is the apparent CA corresponding to the stable equilibrium state, and *r* is the roughness ratio of the material surface, which is the ratio of the true area of the object and affects the homogeneous surface. In addition, *θ* is the Young CA, which represents the CA of the ideal surface. When a voltage or electric field is applied, the droplet may shrink; thus, a more complex model is required to represent the changes in the CA; these changes are illustrated using the CB model (Figure 2b) and written in Equation (3):(3)$$\cos\left(\theta^{*}\right)=r_{f}\;\cos\left(\theta_{Y}\right)+f-1,$$
where *r_f_* is the roughness ratio of the wet surface area, and *f* is the proportion of the solid surface area that is wet by the liquid. In this equation, *r_f_* is equal to *r* when *f* is 1, and the CB equations would become the Wenzel equation. In addition, each fraction of the total surface area is represented by *f_i_* in the presence of various degrees of surface roughness. The sum of all *f_i_* equals 1, or the total surface, and the CB equation can be also written as Equation (4):(4)$$\gamma \cos\left(\theta^{*}\right)=\sum_{n=1}^{N}f_{i}\;\left(\gamma_{i,\mathrm{SV}}-\gamma_{i,\mathrm{SL}}\right),$$
where *γ* is the CB surface tension between the liquid–vapor interface, and *γ_i,_*_SG_ and *γ_i,_*_SL_ are the surface tension between the solid–vapor and solid–liquid interface in every component, respectively.

When a liquid is placed on a substrate, small air pockets are created underneath it; this phenomenon can be described by two models and written as Equation (5) [30]:(5)$$\gamma \cos\left(\theta^{*}\right)=f_{1}\;\left(\gamma_{i,\mathrm{SV}}-\gamma_{i,\mathrm{SL}}\right)-\left(1-f_{1}\right)\gamma$$

In this case, no other surface tension is present between the solid and vapor surface because the air was exposed under the droplet and was the only substrate. Thus, the equation is then expressed as (1 − *f*). The experimental results of the Wenzel and CB models indicated that a Young angle between 90° and 180° can be classified under the CB model because the liquid and air composite system was hydrophobic. The model shifts to the Wenzel model when the droplet wets the surface and does not exceed the edge of the droplet.

## 3. Materials and Experimental Setup

### 3.1. Materials

ITO is composed of indium, tin, and oxygen in different proportions. ITO is colorless and transparent in thin film conditions but yellowish to grey in bulk form. It has favorable electrical conductivity and optical properties (e.g., transparency) and is generally considered a substrate because it is simple to deposit on glass as a film. ITO film is typically deposited on the surface of a substrate by physical vapor deposition, such as sputter deposition or electron beam evaporation techniques [34]. However, optical transparency and electrical conductivity must be carefully balanced to avoid compromising one or the other. When the thickness of ITO film increases, the electrical conductivity also increases because of the greater concentration of charge carriers; however, this reduces the optical transparency. Because of the aforementioned characteristics and advantages, ITO material has been widely applied in flat panel displays, liquid crystal displays, and touch screens (e.g., mobile phones or human–machine interface); it has also been employed in glass- or polymer-based electronics in biochips and bioengineering applications.

### 3.2. Experimental Setup

#### 3.2.1. Pattern Fabrication: UV Laser (Wavelength- 355 nm) Processing

A diode-pumped solid-state UV laser (Coherent, Inc. AVIA 355-14^TM^, Santa Clara, CA, USA) with a wavelength of 355 nm was employed as the laser source (maximum average output power: 14 W; pulse width: 32 ns at a pulse repetition rate of 40 kHz; pulse repetition frequency range: 1–400 kHz). The laser source employed the TEM^00^ transverse eletromagnetic (TEM) mode, with an output beam diameter of 3.5 mm, and the beam quality factor M^2^ was less than 1.3. The beam diameter was magnified by a 2× beam expander. Two-axis high-speed galvanometric scanning mirrors were used to adjust the focus position of the laser beam for the laser direct writing process, and an F-theta lens with a focal length of 160 mm was used to focus the laser light spot on the working plane (Figure 3). The theoretical and actual diameters of the focused laser spot on the working plane were approximately 13.5 and 30 μm, respectively.

First, we used the UV laser source to etch microstructures on the surface of ITO glass. The designed comb patterns had the same line width and line pitches. Microstructures with an ablated line width and pitch of 70, 90, 110, and 110 µm were designed. The four patterns are illustrated in Figure 4.

The maximum output power of the laser system was set at 14 W, 30% of which was employed in the laser ablation process, and the other 70% of power was converted to heat energy to act on the sample surface and escape into the air. The pulse repetition frequency speed was fixed at 100 kHz. The scanning speed was adjusted and set at 500, 750, 1000, 1250, and 1500 mm/s, and the spot overlap (O_R_) was 83.3%, 75%, 66.7%, 58.3%, and 50%, respectively. The number of processing repetitions was set at 1, 3, and 5 to investigate the depth and line roughness of the ablated lines.

#### 3.2.2. Surface Morphology Observation and CA Measurement

After fabrication of the microstructures on the ITO glass (Ruilong Glass, specification with coating thickness 250 nm and 100 ohm/sq) by employing the laser direct writing process, we observed and measured the surface morphology and roughness using a 3D confocal laser scanning microscope (KEYENCE, VK-X200, Itasca, IL, USA) and performed analyses using its software program (KEYENCE, VK-Analyzer Plus). In addition, the CA of the droplet on the surface of the ITO glass, which relates to the wettability, was examined using the developed image system measurement. The deionized water was the liquid employed to measure the CA. A digital microscope was employed to acquire the droplet image, and the CA was measured and analyzed from the acquired image. Therefore, the calibration of the measured angle from the acquired image was crucial and provided reference information for the subsequent experiment. The block gauge was designed with an angle of 30°, 60°, and 90°. After the calibration process, the droplet was placed on the ITO glass to acquire the droplet image by using the digital microscope; subsequently, the CA was measured using the droplet image in ImageJ (National Institute of Health, Bethesda, MD, USA).

#### 3.2.3. Experimental Process

Five primary steps were involved in producing the microstructures on the ITO glass and obtaining the CA to determine the wettability of the ITO glass surface; the adjusted and measured target of each step were as follows:Step (I):AutoCAD (2017, Autodesk, San Rafael, CA, USA) was used to design the writing path of the laser spot. The left line and ablated line width of the ITO film were the same dimension, either 70, 90, 110, or 130 μm.Step (II):The parameters, such as the power and pulse repetition frequency of the laser system, were fixed, and the scanning speed and repetition times were adjusted to generate the comb-pattern microstructures with varying surface roughness and depth.Step (III):A 3D confocal microscope was used to measure the edge contour, line depth, and the morphology of the microstructures, and we analyzed the relationship of the depth and surface roughness of the ablated lines with the various repetition times and scanning speeds during laser treatment. The surface roughness is reported as mean height (*S*a) and root mean square height (*S*q). The *S*a and *S*q were defined by the average value and root mean square along the sampling area, respectively. In addition, the roughness of ablated line was also measured and defined as *R*a (mean height) and *R*q (root mean square).Step (IV):A digital microscope was set up to acquire a cross-section image of the droplet. Before acquiring the droplet image, we used the block gauge with an angle of 30°, 60°, and 90° to acquire the image and analyze the angle for the calibration process (Figure 5).Step (V):After the calibration, we placed the ITO glass on the stage in front of the digital microscope and added a droplet of deionized water on the microstructure region of the ITO glass surface. Each experimental parameter was repeated three times, and the volume of the droplet was determined in the first experiment. In addition, an electrical charge was exerted on the pad to observe the deformation of the droplet and investigate the variation in the CA. The relationship between the CA, surface roughness, ablated line pitch, and depth of the ITO film was analyzed to obtain the optimal parameters of the microstructure for enhancing the CA and generating a spherical droplet for biochip application.

## 4. Experimental Results and Discussion

To investigate the effect of droplet volume, ablated pattern dimension, and surface properties on the wettability of a droplet on an ITO substrate by laser treatment, we analyzed the surface roughness of the ablated pattern and the CA of the droplet. The supporting angle of the droplet is affected by the ablated lines when the droplet is placed on the ITO substrate; the droplet is supported on or falls into the ablated lines according to the dimensions of the lines and the droplet volume. Therefore, the measured CA was divided into the values measured from the direction parallel (side view) and perpendicular (front view) to the ablated pattern (Figure 6). The surface roughness of the ablated pattern was the average value of the full area, which was approximately 1.6 × 1.0 mm^2^ under the 10× objective lens. In addition, the line roughness was the average value of the area (90 × 90 μm^2^).

### 4.1. Effect of Droplet Volume on CA

A liquid’s volume determines the gravitational force it exerts, and the effect of droplet volume on CA should be minimized in biochips. Therefore, we measured the CA of droplets with different volumes on flat ITO glass to determine the optimal droplet volume for the subsequent experiment. On the original surface of the ITO substrate, the CA of the droplet was affected by the surface properties of the ITO film and the droplet’s gravitational force. As the volume of the droplet increased, a larger area was required to support it. The experiment results revealed that the CA of the droplet on the flat ITO substrate remained in the range of 91° ± 2° when employing a liquid volume of 5 to 20 μL. However, the CA was affected when droplets with various volumes were placed on the ablated pattern with the 110 μm line width and pitch. With a small liquid volume, the droplet was strongly affected by the surface morphology and infiltrated into the valley of the ablated lines; thus, the CA decreased evidently with the decrease in droplet volume. This was particularly evident in the front-view analysis (Figure 7). The edge of the droplet was caught by each ablated line, which resulted in the lower CA when the droplet was placed on the ITO substrate with the laser-ablated microstructure and observed from the perpendicular direction. In addition, the CA was only affected by the outermost microstructure when the CA droplet was observed from a direction parallel to the ablation lines; thus, the CA from the side view was slightly higher than that observed from the front view. To prevent the scale effect on the CA of the droplet, the liquid volume was set at 20 μL for the following experiment because this volume exhibited the smallest variation of CA between the flat substrate and the ablated pattern.

### 4.2. Effect of Scanning Speed on CA and Line Roughness

After the droplet volume was determined, we employed the 3D confocal laser scanning microscope to observe and analyze the line roughness of the ablated pattern produced by the laser system. The width of the ablated line and the left width was 110 μm, and five laser-spot scanning speeds were employed to ablate the ITO film. The optical image and the analysis results (Figure 8 and Table 1) revealed that the line roughness of the ablated pattern can be reduced using a slower scanning speed because it has high spot overlap. With a scanning speed of 1500 mm/s or higher, the ablated region became shattered and discontinuous, and the line roughness obtained with a scanning speed of 1500 mm/s was nearly twice that of 500 and 750 mm/s. In this situation, the droplet would infiltrate into the microstructure when the surface is slightly rough and the depth is small, such as in the Wenzel state; the droplet image revealed that the CA decreased with the increase in scanning speed and line roughness (Figure 9). Therefore, the optimal scanning speed for laser ablation of the ITO substrate ranged from 500 to 1000 mm/s, with the most favorable results at 750 mm/s. The surface of the microstructures fabricated with a laser scanning speed of 750 mm/s had the smallest variation in CA between the side view and front view and in the same view than the other microstructures in the current experiments. The results also indicated that the variation between the two view directions and self-variation in the other experiments increased with the increase of scanning speed.

### 4.3. Effect of Line Pitch and Laser Scanning Speed on CA

After we determined the optimal laser scanning speed, we analyzed the effect of ablated line width and pitch by changing the pattern dimensions. The pattern was fabricated using different line width, line pitch, and scanning speed parameters; subsequently, the actual ablation line dimension was measured in the same observation area using a 10× objective lens under the 3D laser scanning microscope (Figure 10). The results indicated that the actual total ablated line widths were approximately 584, 570, 555, and 567 μm with the design pattern dimensions of 70, 90, 110, and 130 μm, respectively. Because the design line width and pitch of 110 μm resulted in a lower ablation region than that in designs using other dimensions, a lower surface roughness *S*a (0.518 μm) and *S*q (0.660 μm) were simultaneously obtained (Table 2).

For most scanning speeds, the 110-μm pattern exhibited a higher CA than the other line widths and pitches (Figure 11). In addition, we maintained a fixed scanning speed and examined the scale effect on CA. The surface roughness *S*a and *S*q increased from 0.518 to 0.679 μm and from 0.66 to 0.846 μm, respectively, when the line width and pitch increased from 110 to 130 μm. In the construction of the pattern, two or three laser ablation paths overlapped on the wider ablation line; thus, a higher surface roughness was also observed. The CA decreased evidently when the droplet was dropped on the ITO substrate with the 130-μm microstructure and infiltrated into the microstructure, especially from the front direction and with a 1500-mm/s scanning speed. The edge of the droplet was evidently caught by the microstructure and rough surface on the ITO substrate when the scanning speed was 1500 mm/s. Therefore, the CA from the front view decreased to approximately 50° for the pattern line width of 130 μm and scanning speed higher than 1000 mm/s (Figure 11c–e). Therefore, the pattern line width of 110 μm, which had the smallest ablation width and surface roughness, exhibited the highest droplet CA, which was closest to the value on a flat ITO surface. Moreover, the optimal scanning speed was less than 1000 mm/s, with 750 mm/s being ideal. The microstructure with a line width of 110 μm fabricated using a laser scanning speed of 750 mm/s had a sideview and front view CA of 83° ± 1° and 78.5° ± 2.5°, respectively.

### 4.4. Effect of Repetition Times on CA

On the basis of the aforementioned results, the line width and pitch dimension of 110 μm was selected and the laser scanning speed of 750 mm/s was employed to fabricate the microstructures. However, the depth of the microstructures also affects the droplet’s CA because of the supporting force and infiltration state; thus, we adjusted the repetition times of the ablation process to fabricate microstructures with various depths and analyzed the CA of the droplet on these patterns. As the depth increased, the surface roughness of the ablation line decreased because the thermal energy flattened these microstructures when the laser ablation repetition times increased (Figure 12). The depth of the ablated pattern was approximately 0.13, 0.2, and 0.45 μm with 1, 3, and 5 repetition times, respectively. In addition, the line roughness *R*a was 0.557, 0.353, and 0.325 μm, and the *R*q was 0.794, 0.498, and 0.451 μm, respectively. The CA of the droplet decreased when the repetition times increased from 1 time to 3 times, which may have been caused by the droplet infiltrating into the deeper microstructures of the ablated pattern (Figure 13). However, the CA slightly increased when the repetition times increased from 3 times to 5 times because of the smaller line roughness of 5 times ablation than 3 times ablation caused by the thermal effect. The variation in the droplet’s CA also increased with the increase in repetition times because the edge of the droplet was easily caught by the contour of the ablation line in deeper microstructures. Therefore, the repetition time of the ablation process was fixed at one time; in this manner, the CA of the droplet can be maintained at approximately 80° and 76° from the side and front view, respectively.

### 4.5. Effect of Electric Field on CA

Because ITO is colorless and has favorable electric conductivity, it is widely used as a substrate in cases where an electric field must be applied to observe the change in sample characteristics. In the current experiment, an electric field was applied on the positive and negative pad to generate a staggered field on the droplet; the current was fixed at 100 mA, and the voltage (ranging from 1 to 7 V) was adjusted to observe the variation in the droplet’s CA. To investigate the effect of an electric field on the droplet, the CA before and after applying the voltage was measured, and the application time was initially fixed at 5 s. The results revealed that the initial values from the sideview before applying the voltage were consistent, which indicates the stability of the system and the repeatability. The variation in the CA from the sideview between the period before and after the voltage application decreased as the voltage increased, and it had the smallest variation at 5 V (Figure 14). The CA of the droplet from the front view also decreased because of the charge accumulation while the voltage was applied. As the voltage increased, the droplet rapidly electrolyzed, and the variation in the CA increased again. In addition, the variation in the CA between the period before and after voltage application decreased because both the left and right edge of the droplet step cross several ablation lines from the front view, and the effect of the electric field was lower than that in the droplet measured from the sideview that may switch the occupy position by the electric field.

After we determined the voltage application limit, we investigated the optimal voltage application time for maintaining the droplet’s characteristics. The application voltage was fixed at 5 V, and four application times (ranging from 5 s to 30 s) were employed to observe the change in the droplet’s CA. The results revealed that the CA before and after applying the voltage varied the least when the application time was under 10 s, regardless of the viewing direction. The variation in the CA before and after the voltage application and the self-measured bias of the CA increased when the application time exceeded 20 s (Figure 15). This phenomenon was attributed to charge accumulation on the droplet and even electrolysis when the application time increased to 20 or 30 s. Therefore, the optimal time for voltage application was less than 20 s, preferably less than 10 s, to maintain the droplet profile and prevent it from becoming hydrophilic.

The current results elucidated the ideal dimensions of the comb pattern, as well as the optimal parameters for the laser ablation process, on an ITO substrate. This information can guide the fabrication of microstructures that help the surface retain its neutral status, instead of becoming hydrophilic, for biomedical applications. Moreover, the voltage application parameters were determined, and the droplet morphology and electric or optical effect on the ITO substrate were observed to serve as a reference for charging a droplet in a biomedical chip.

## 5. Conclusions

To maintain the wettability of droplets on ITO substrates with microstructures for various applications, we applied laser treatment in this study to fabricate a comb pattern and investigate the relationship between the CA and the line width and pitch of the pattern, the surface roughness, and the depth of the ablated trace with various laser scanning speeds and repetition times. The results revealed that the CA and spherical droplet morphology can be maintained when the surface roughness is decreased by reducing the laser scanning speed. We obtained the most neutral surface, with a droplet CA similar to that on a flat ITO surface, at a laser scanning speed of 750 mm/s, and the line roughness *R*a and *R*q were <0.3 and 0.5 μm, respectively. In addition, a line width and pitch of 130 μm with a scanning speed of 750–1500 mm/s generated a hydrophilic surface; thus, the slightly hydrophilic and nearly neutral surface could be produced with a line width and pitch of 110 μm and a laser scanning speed of 750 mm/s, which resulted in a CA of 83° ± 1° and 78.5° ± 2.5° from the side and front view, respectively. In addition, an examination of the droplet’s electrical resistance revealed that it has the smallest CA variation when a voltage of 5 V is applied for fewer than 10 s. In the future, the ITO substrate can be treated to fabricate electric pads for biomedical and optoelectrical applications, and the wettability of a biomedical liquid sample can be controlled by adjusting the pattern dimension and laser processing parameters. Different patterns can also be directly fabricated using a laser system. Such a system is fast, offers high pattern design flexibility, and enables the adjustment of the CA. Moreover, ITO substrate has high electrical conductivity and optical penetration characteristics, making it suitable for microfluidic or cell manipulation in biomedical applications. The laser-based processing method also can be used to mill the substrate and analyze the surface properties and effect for fabricating the chip in other further application in the future.

## Figures and Tables

**Figure 1 micromachines-12-00044-f001:**
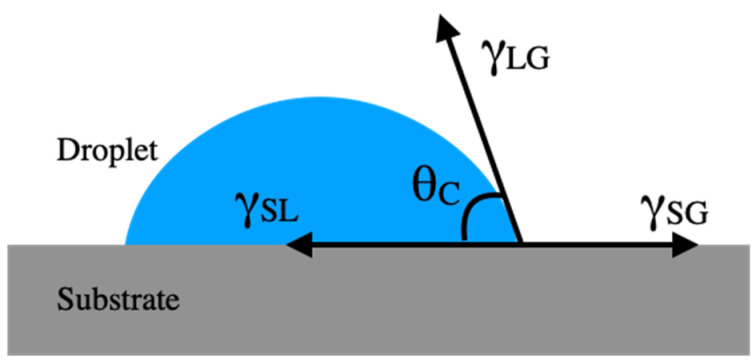
Schematic of droplet equilibrium on a substrate according to Young’s equation. *γ*_SG,_ solid–vapor interfacial energy; *γ*_SL,_ solid–liquid interfacial energy; *γ*_LG,_ liquid–vapor interfacial energy.

**Figure 2 micromachines-12-00044-f002:**
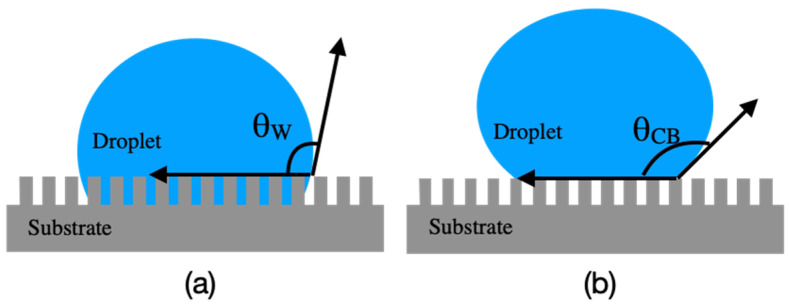
Droplet on microstructures with (**a**) Wenzel (W) state, and (**b**) Cassie–Baxter (CB) state. *θ*, Young contact angle (CA).

**Figure 3 micromachines-12-00044-f003:**
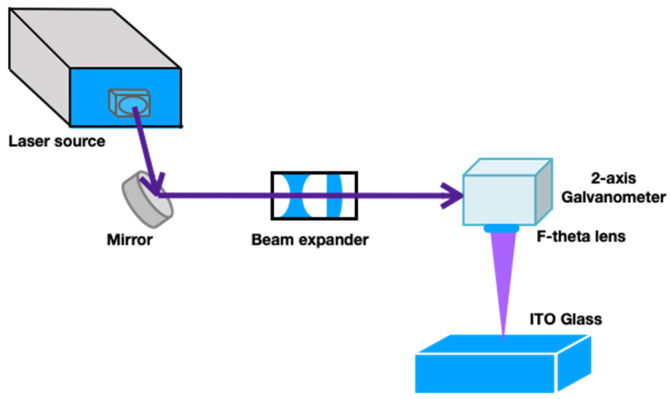
Laser direct writing system setup on indium tin oxide (ITO) glass.

**Figure 4 micromachines-12-00044-f004:**
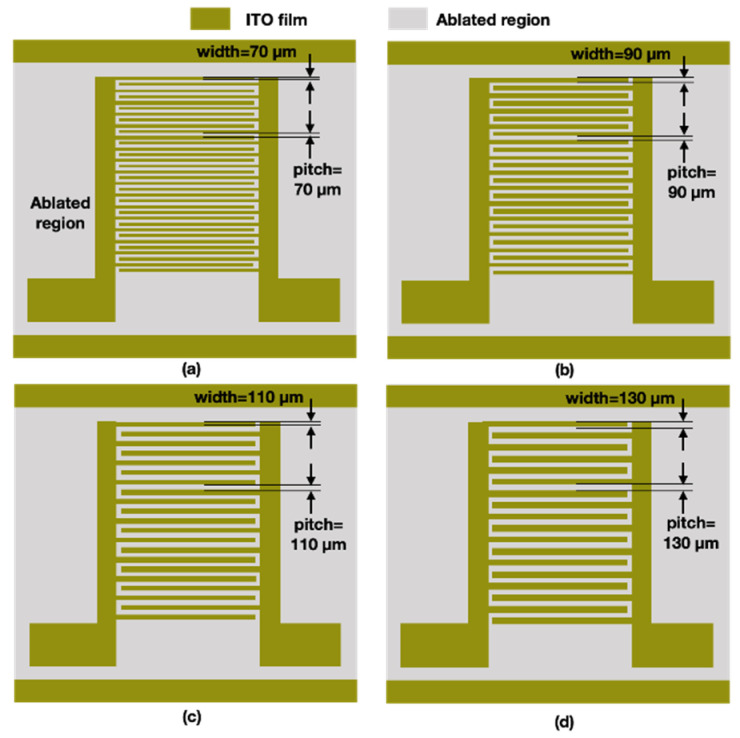
Schematic of comb-pattern microstructures with four different line pitches and widths: (**a**) 70 μm, (**b**) 90 μm, (**c**) 110 μm, and (**d**) 130 μm. ITO, indium tin oxide.

**Figure 5 micromachines-12-00044-f005:**
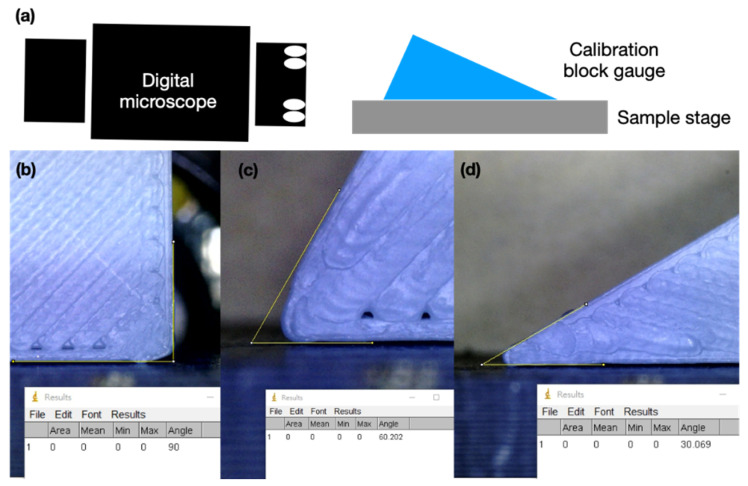
Schematic of (**a**) calibration of the digital microscope by using a block gauge with (**b**) 90°, (**c**) 60°, and (**d**) 30°.

**Figure 6 micromachines-12-00044-f006:**
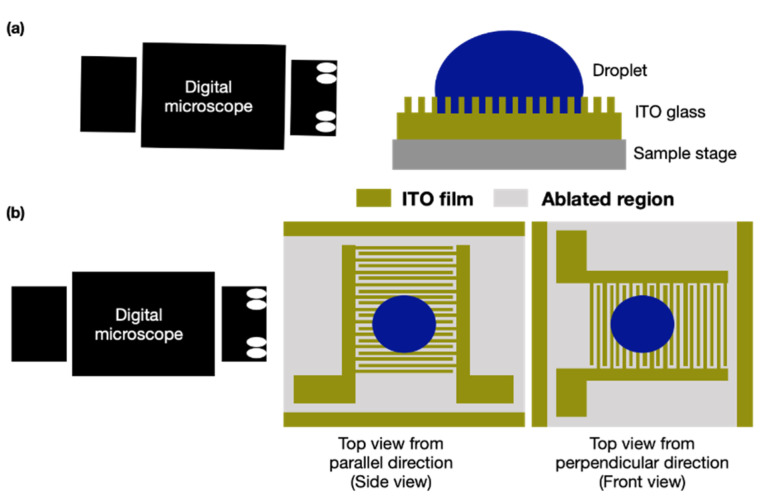
Acquired image and analysis of droplet angle through (**a**) cross section from front view, and the schematic of the top view for observing the droplet by (**b**) side view and front view on an indium tin oxide (ITO) substrate with an ablated pattern.

**Figure 7 micromachines-12-00044-f007:**
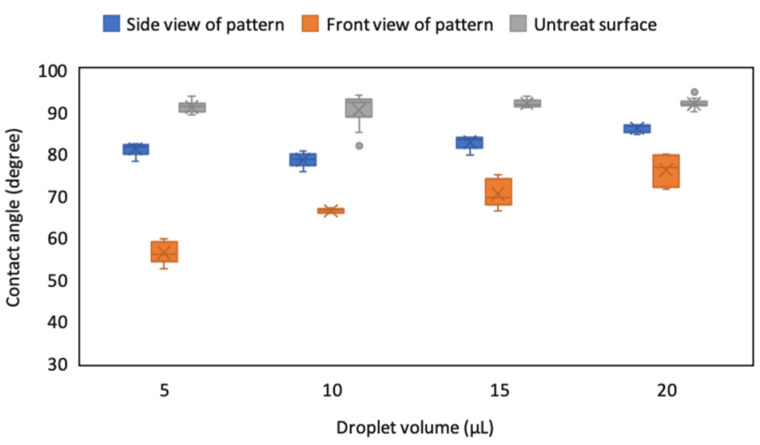
Contact angle (CA) of droplets with various volumes on the ITO substrate and the ablated pattern with a dimension of 110 μm.

**Figure 8 micromachines-12-00044-f008:**
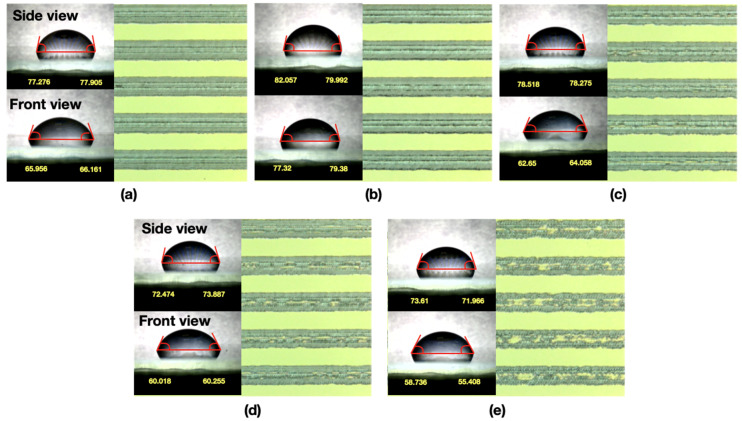
Optical images of the ITO substrate ablated by the laser system with line width and pitch of 110 μm and scanning speeds of (**a**) 500, (**b**) 750, (**c**) 1000, (**d**) 1250, and (**e**) 1500 mm/s.

**Figure 9 micromachines-12-00044-f009:**
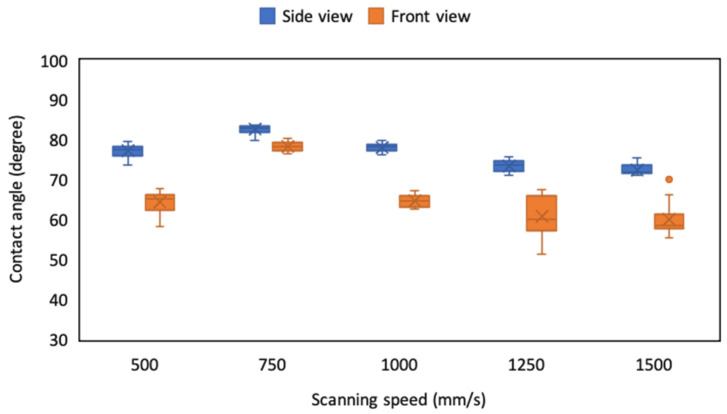
CA of droplets on the ITO-substrate ablated pattern with a dimension of 110 μm and various scanning speeds.

**Figure 10 micromachines-12-00044-f010:**
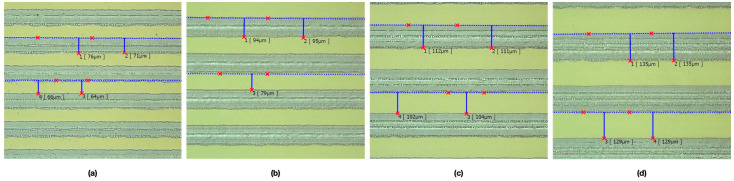
Optical images of the ITO substrate ablated by the laser system with a scanning speed of 750 mm/s and a line width and pitch of (**a**) 70, (**b**) 90, (**c**) 110, and (**d**) 130 μm.

**Figure 11 micromachines-12-00044-f011:**
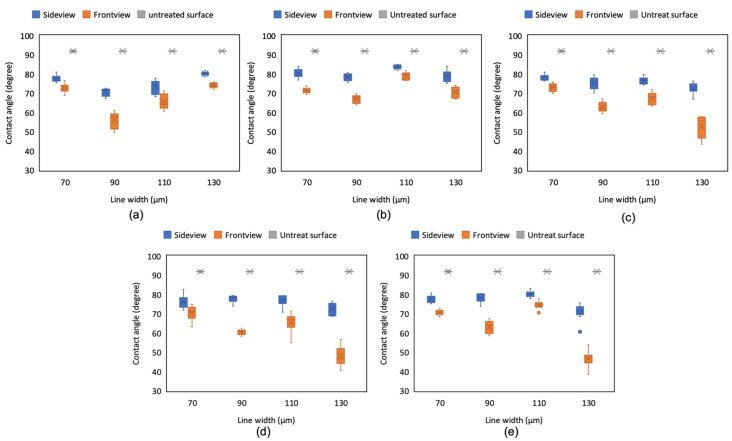
CA of droplets on the ITO substrate from the side and front view of the ablated pattern with various line pitches and widths and scanning speeds of (**a**) 500, (**b**) 750, (**c**) 1000, (**d**) 1250, and (**e**) 1500 mm/s.

**Figure 12 micromachines-12-00044-f012:**
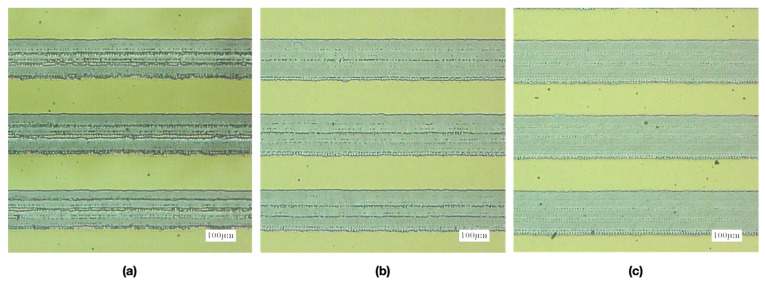
Optical images of the ITO substrate with line width and pitch of 110 μm ablated by the laser system with a scanning speed of 750 mm/s and repetition times of (**a**) 1, (**b**) 3, and (**c**) 5.

**Figure 13 micromachines-12-00044-f013:**
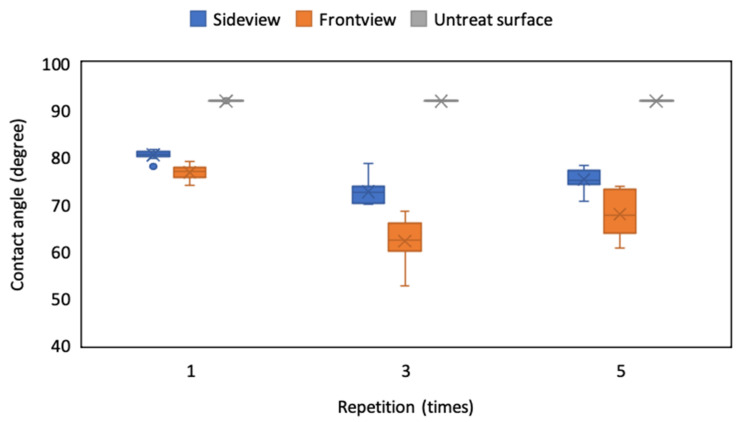
CA of droplets on the ITO substrate from the side and front view of the ablated pattern with different repetition times.

**Figure 14 micromachines-12-00044-f014:**
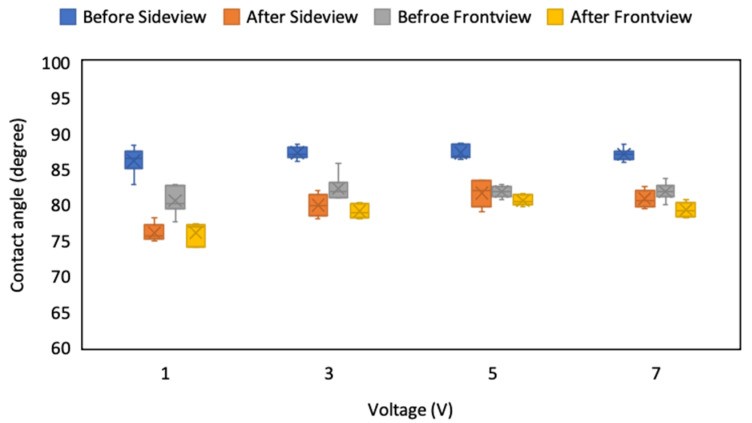
CA of droplets on the ITO substrate from the side and front view of the ablated pattern before and after applying different voltages.

**Figure 15 micromachines-12-00044-f015:**
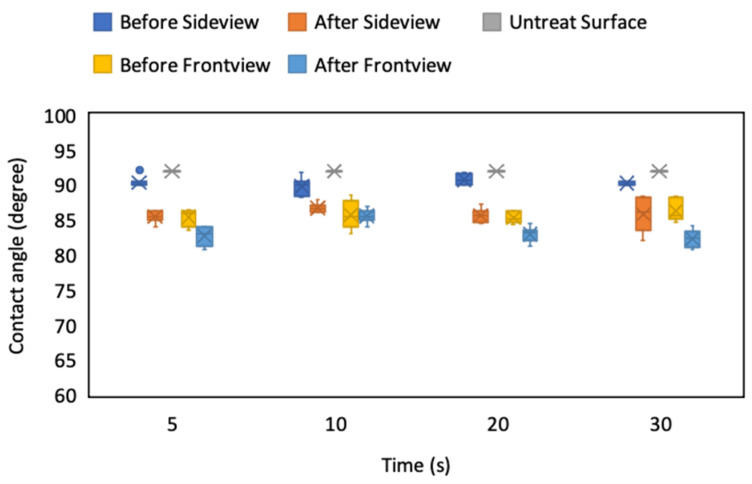
CA of droplets on the ITO substrate from the side and front view of the ablated pattern before and after applying 5 V for various periods.

**Table 1 micromachines-12-00044-t001:** Line roughness *R*a (mean height) and *R*q (root mean square) of ablated lines on the indium tin oxide (ITO) substrate treated by different laser scanning speeds.

Scanning Speed (mm/s)	500	750	1000	1250	1500
*R*a (μm)	0.292	0.290	0.453	0.493	0.573
*R*q (μm)	0.399	0.411	0.619	0.652	0.741

**Table 2 micromachines-12-00044-t002:** Surface roughness *S*a (mean height) and *S*q (root mean square) of ablated lines on the ITO substrate fabricated by the laser scheme 750 mm/s and different widths and pitches.

Line Width and Pitch (μm)	70	90	110	130
*S*a (μm)	0.635	0.632	0.518	0.679
*S*q (μm)	0.793	0.788	0.660	0.846

Notes: *S*a, mean height; *S*q, root mean square.

## Data Availability

The data has never been published before and the original source.

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
