# Peer review of "Surface Wettability and Electrical Resistance Analysis of Droplets on Indium-Tin-Oxide Glass Fabricated Using an Ultraviolet Laser System"

_micromachines, 2021, doi:10.3390/mi12010044_

Round 1

Reviewer 1 Report

The authors present an interesting study on the wettability of ITO after laser trattament. The topic is interesting and the results well reported. The major issue concerns the aim of the work. It is not clear if you want or not to modify the wettability of ITO and if you need a hydrophobic surface or not, and why.

In particular the abstract and  the introduction should be revised in order to make clear your scope.

Some minor issues are reported below:

L 12: what do you mean with "normal"?

L 11-13. Here you write that you want modify the hydrophilic behaviour of the ITO for biochip applications, but in the results you find the combination of laser paremeter that give the CA more similar to that of non-textured sample. This point should be clarified. Moreover,  It could be helpful for the reader making explicit the wetting behaviour you want to reproduce (hydrophobic, superhydrophilic..?) and why.

L25: "and resulted in the droplet becoming hydrophilic." A droplet can not be hydrophilic, a surface can be hydrophiilc! Moreover, a surface is hydrophilic yet at CA<90°. So, it was already hydrophilic at a laser scanning speed of 750 mm/s.

L 30: Transparent: Are you sure that After laser fabrication the transparency of the chip is preserved?

L39-41 These lines deserve some citations, almost one for each application, Some interesting and recent review on these topics are: DOI: 10.1039/c8an01061g, as DOI 10.1002/elps.201800361, DOI  10.3390/mi10090594

L 48 Ref 2 is referred to glass chip, not to polymeric ones. I could suggest a very recent work: https://doi.org/10.1016/j.eng.2020.10.012.

L55: These properties are also of PDMS, not just of glasses.

L58: What do you mean with "conventional method"?

L58-60: Here, you repeat the properties of glass already mentioned in L55-56

L62: are you sure that a process of 12 minutes is ready for mass scale production?

L55-65: A ref to a glass chip fabricated by laser could be added (e.g. DOI: 10.5772/64324)

L128: What do you mean with normal? A contact angle renge should be make explicit.

In all the introduction it is not clear what do you want to obtain on your surface: hydrophilicity? hydrophocibicity? Why hydrophobicity and not superhydrophobicity? 

L 160 Wenzel

L 163 Wenzel

Fig 2: Normal state is not a measure of the wettability of a surface

L 192 193 These lines deserve a citation.

Section 3.1. Here you describe the general  properties of ITO, but don't provide the technical specifications and brand of the ITO you used.

L229: How do you use the other 70%?

In the label you should Write if Fig 6 (a) refers to the cross section of front view or side view

Fig 8: Please, indicate with a label the CA for each droplet figure.

Fig 10 enlarge the measure bar on the figures

Fig 11: Does the X axis represent the pitch or the width?

In figs 11, 13 and 15 the CA of the untreated surface should be inserted, for example with a dotted line.

Fig 13 and 15: enlarge the y axes.

L 432-433 The surface is hydrophilic (or not), not the droplet!

Line 455: How your study can help in the chip fabrication? To fabricate a chip you must mill the substrate, not just tracking some lines on it. 

Author Response

Response to Editor and Reviewers

Manuscript# micromachines-1040975

Title: Surface Wettability and Electrical Resistance Analysis of Droplet on Indium-Tin-Oxide Glass Fabricated Using an Ultraviolet Laser System

Authors: Hsin-Yi Tsai, Chih-Ning Hsu, Cheng-Ru Li, Yu-Hsuan Lin, Wen-Tse Hsiao, Kuo-Cheng Huang, J. Andrew Yeh

Dear Editor,

   Thank you very much for yor suggestions. Enclosed the mail, please find one copy of the revised manuscript of the above-mentioned paper which has been carefully prepared following your suggestions.

   The authors deeply appreciate the referees for their valuable comments. We modify the manuscript accordingly, and the detailedcorrections are listed below point by point:

Reviewer’s comment,

Reviewer#1

The authors present an interesting study on the wettability of ITO after laser trattament. The topic is interesting and the results well reported. The major issue concerns the aim of the work. It is not clear if you want or not to modify the wettability of ITO and if you need a hydrophobic surface or not, and why.

In particular the abstract and the introduction should be revised in order to make clear your scope.

Response: Thanks for the reviewer’s suggestion. In the revised paper, the aim of the work was added in abstract and the introduction of mauscript. Therein, the wettability of substrate was important to the biomedical chip application. When the surface was a normal or hydrophobic surface, the droplet was easily to be moved and can be manipulated to the specific region on the chip. Therefore, we aim to fabricate the microstructure on the ITO glass to discuss the relationship of pattern dimension and contact angle and investigate the wettability and electric resistance of droplet for future biomedical application. With these information, the suitable dimension and processing parameter of microstructures and the limit of employed electric voltage were obtained, and the operating parameters of droplet and reagent can be determined to prevent the droplet attached on the chip’s surface.

Some minor issues are reported below:

  • L 12: what do you mean with "normal"?

Response: Thanks for the reviewer’s suggestion, in this manuscript the droplet with contact angle of approximately 90° was defined as normal.

  • L 11-13. Here you write that you want modify the hydrophilic behaviour of the ITO for biochip applications, but in the results you find the combination of laser paremeter that give the CA more similar to that of non-textured sample. This point should be clarified. Moreover, It could be helpful for the reader making explicit the wetting behaviour you want to reproduce (hydrophobic, superhydrophilic..?) and why.

Response: Thanks for the reviewer’s suggestion. In the initial purpose, the surface properties of ITO glass was modified into hydrophobic for droplet and reagent manipulation in biomedical application. However, the experimental results indicated that the designed pattern of microstructures produce the hydrophilic surface instead of hydrophobic surface. Therefore, the discussion and the conclusion were modified into the less variation of CA caused by dimension and processing parameters of laser treatment. Without the laser treatment on ITO glass, the substrate was fully conductive and was difficult to manipulate the droplet Therefore, the wetting behavior of hydrophobic or nearly hydrophobic surface was preferred to reproduce to let the droplet become a spherical morphology for manipulation and sensing in biomedical application.

  • L25: "and resulted in the droplet becoming hydrophilic." A droplet can not be hydrophilic, a surface can be hydrophiilc! Moreover, a surface is hydrophilic yet at CA<90°. So, it was already hydrophilic at a laser scanning speed of 750 mm/s.

Response: Thanks for the reviewer’s suggestion. The droplet was modified into the surface that can be hydrophilic. The surface was hydrophilic at a laser treatment with scanning speed of 750 mm/s, and the dimension and processing parameters that can cause the less variation of CA was discussed and investigated the let the droplet try to be spherical morphology for droplet manipulation.

  • L 30: Transparent: Are you sure that After laser fabrication the transparency of the chip is preserved?

Response: Thanks for the reviewer’s suggestion. The description of transparent was modified the light transmittance in the revised manuscript because the surface will be rough after laser ablation.

  • L39-41 These lines deserve some citations, almost one for each application, Some interesting and recent review on these topics are: DOI: 10.1039/c8an01061g, as DOI 10.1002/elps.201800361, DOI  10.3390/mi10090594

Response: Thanks for the reviewer’s suggestion. The suggestion references were added in the Introduction in the revised manuscript.

  • L 48 Ref 2 is referred to glass chip, not to polymeric ones. I could suggest a very recent work: https://doi.org/10.1016/j.eng.2020.10.012.

Response: Thanks for the reviewer’s suggestion. The suggestion reference was added in the Introduction in the revised manuscript.

  • L55: These properties are also of PDMS, not just of glasses.

Response: Thanks for the reviewer’s suggestion, the PDMS was added in the revised manuscript.

  • L58: What do you mean with "conventional method"?

Response: Thanks for the reviewer’s suggestion, the conventional method indicated that the pattern determined by the mask and produced by scanning electron microscopy or focused ion beam technology.

  • L58-60: Here, you repeat the properties of glass already mentioned in L55-56

Response: Thanks for the reviewer’s suggestion, the repeated properties of glass was removed in the revised manuscript.

  • L62: are you sure that a process of 12 minutes is ready for mass scale production?

Response: Thanks for the reviewer’s suggestion, the needed time of 12 minute was from the reference and may be we can remove the time information for the production because we are not sure the scale.

  • L55-65: A ref to a glass chip fabricated by laser could be added (e.g. DOI: 10.5772/64324)

Response: Thanks for the reviewer’s suggestion. The suggestion reference was added in the Introduction in the revised manuscript.

  • L128: What do you mean with normal? A contact angle range should be make explicit.

Response: Thanks for the reviewer’s suggestion, the droplet with contact angle of approximately 90° was defined as normal.

  • In all the introduction it is not clear what do you want to obtain on your surface: hydrophilicity? hydrophocibicity? Why hydrophobicity and not superhydrophobicity? 

Response: Thanks for the reviewer’s suggestion, we added the description of the hydrophobic surface was that we want because the droplet and reagent on the microstructures can be easily manipulated. Howerever, the droplet was difficult to stand on the substrate with superhydrophobic surface for the further biomedical reaction.

  • L 160 Wenzel

Response: Thanks for the reviewer’s suggestion, the Wentzel was modified into Wenzel in the revised manuscript.

  • L 163 Wenzel

Response: Thanks for the reviewer’s suggestion, the Wentzel was modified into Wenzel in the revised manuscript.

  • Fig 2: Normal state is not a measure of the wettability of a surface

Response: Thanks for the reviewer’s suggestion, the normal state was removed in Fig. 2 in the revised manuscript.

  • L 192 193 These lines deserve a citation.

Response: Thanks for the reviewer’s suggestion, the citation was added into these lines in the Section 2 of revised manuscript.

  • Section 3.1. Here you describe the general properties of ITO, but don't provide the technical specifications and brand of the ITO you used.

Response: Thanks for the reviewer’s suggestion, the technical specification and the brand of ITO glass was added in the Section 3.2.2 in the revised manuscript.

  • L229: How do you use the other 70%?

Response: Thanks for the reviewer’s suggestion, the other 70% of power was converted to heat energy to act on the sample surface and escape into the air. The description was added in the Section 3.2.1 of revised manuscript.

  • In the label you should Write if Fig 6 (a) refers to the cross section of front view or side view

Response: Thanks for the reviewer’s suggestion, the description of Fig. 6(a) was added in the revised manuscript, which refers the cross section of front view.

  • Fig 8: Please, indicate with a label the CA for each droplet figure.

Response: Thanks for the reviewer’s suggestion, the CA of each droplet was added in Fig. 8 in the revised manuscript.

  • Fig 10 enlarge the measure bar on the figures

Response: Thanks for the reviewer’s suggestion, the measure bar on figures were enlarged in the Fig. 10 in the revised manuscript.

  • Fig 11: Does the X axis represent the pitch or the width?

Response: Thanks for the reviewer’s suggestion, the X axis in Fig. 11 represent the line width, and it was modified in the Fig. 11 in the revised manuscript.

  • In figs 11, 13 and 15 the CA of the untreated surface should be inserted, for example with a dotted line.

Response: Thanks for the reviewer’s suggestion, the CA of untreated suface was added in figs 11, 13 and 15, please find the revised manuscript.

  • Fig 13 and 15: enlarge the y axes.

Response: Thanks for the reviewer’s suggestion, the y axes in Fig. 13, 14 and 15 were enlarged, please find the revised manuscript.

  • L 432-433 The surface is hydrophilic (or not), not the droplet!

Response: Thanks for the reviewer’s suggestion, the droplet was modified into the surface, please find the revised manuscript.

  • Line 455: How your study can help in the chip fabrication? To fabricate a chip you must mill the substrate, not just tracking some lines on it. 

Response: Thanks for the reviewer’s suggestion, the study provides the information of surface modification parameters and it effect on droplet contact angle by laser system. For further fabrication of chip and other application, the system and method also can be employed to mill the substrate such the glass. The surface properties should be analyzed again after the milling on substrate for the actual application. Please find the description in the conclusion of revised manuscript.

The revised paper has been resubmitted to you, please see the attachment. We are looking forward to your positive response.

Hsin-Yi Tsai

Associate Researcher

Taiwan Instrument Research Institute, National Applied Research Laboratories, Taiwan

Reviewer 2 Report

The manuscript titled »Surface Wettability and Electrical Resistance Analysis of Droplets on Indium-Tin-Oxide Glass Fabricated Using an Ultraviolet Laser System« is an intriguing research contribution offering new insights in the field of micromachines. The authors have used an ultraviolet laser to pattern comb-like structures onto ITO glass. After optimisation of UV laser treatment parameters, CA and resistance analysis were performed.

The manuscript is suitable for publication in the journal Micromachines after minor revision. Here are some comments that should be taken into consideration:

Line 12, “normal or hydrophilic” What is normal? You should explain what normal means in your context. There is an explanation (lines 146-151) but it does not include normal, above 90 degrees (90-94) is defined as hydrophobic. Maybe use the term “neutral” instead of normal.

Line 21, “90°–94°” should be replaced by 92°±2° (like in line 300), same goes for lines 23, 372, 451…

Line 25, “droplet becoming hydrophilic” The surface can become hydrophilic, the droplet forms accordingly but never changes its hydrophilicity.

Line 58, “fewer” should be replaced by less.

Line 111, “on” makes the sentence not understandable. It also makes the following lines (112-115) hard to understand. Please reform.

Line 128, “normal or hydrophobic” Again, same as line 12.

Line 262, Surface/area roughness parameters are Sa, Sq. etc. Line/profile roughness parameters are Ra, Rq, etc. Please note the difference. This error occurs in the manuscript multiple times! (lines 288, 336, 349, 354, 359…)

Line 386, Roughness units missing, please add units.

Line 387, “The CA of the droplet decreased with the increase in repetition times” By increasing the repetition times from 3 to 5 the CA increases, making this statement questionable. Reform.

Line 432, “normal status” Again, same as line 12.

I am also missing the information about the liquid used for measuring CA? This should be added to “3.2.2. Surface morphology observation and CA measurement” section.

Author Response

Response to Editor and Reviewers

Manuscript# micromachines-1040975

Title: Surface Wettability and Electrical Resistance Analysis of Droplet on Indium-Tin-Oxide Glass Fabricated Using an Ultraviolet Laser System

Authors: Hsin-Yi Tsai, Chih-Ning Hsu, Cheng-Ru Li, Yu-Hsuan Lin, Wen-Tse Hsiao, Kuo-Cheng Huang, J. Andrew Yeh

Dear Editor,

   Thank you very much for yor suggestions. Enclosed the mail, please find one copy of the revised manuscript of the above-mentioned paper which has been carefully prepared following your suggestions.

   The authors deeply appreciate the referees for their valuable comments. We modify the manuscript accordingly, and the detailedcorrections are listed below point by point:

Reviewer’s comment,

Reviewer#2

The manuscript titled »Surface Wettability and Electrical Resistance Analysis of Droplets on Indium-Tin-Oxide Glass Fabricated Using an Ultraviolet Laser System« is an intriguing research contribution offering new insights in the field of micromachines. The authors have used an ultraviolet laser to pattern comb-like structures onto ITO glass. After optimisation of UV laser treatment parameters, CA and resistance analysis were performed.

The manuscript is suitable for publication in the journal Micromachines after minor revision. Here are some comments that should be taken into consideration:

  • Line 12, “normal or hydrophilic” What is normal? You should explain what normal means in your context. There is an explanation (lines 146-151) but it does not include normal, above 90 degrees (90-94) is defined as hydrophobic. Maybe use the term “neutral” instead of normal.

Response: Thanks for the reviewer’s suggestion, the droplet with contact angle of approximately 90° was defined as normal, and the normal was replaced by neutral in the revised manuscript.

  • Line 21, “90°–94°” should be replaced by 92°±2° (like in line 300), same goes for lines 23, 372, 451…

Response: Thanks for the reviewer’s suggestion, the CA of “90°–94°” was replaced by 92°±2° in the revised manuscript

  • Line 25, “droplet becoming hydrophilic” The surface can become hydrophilic, the droplet forms accordingly but never changes its hydrophilicity.

Response: Thanks for the reviewer’s suggestion, the droplet was replaced by the surface in the revised manuscript.

  • Line 58, “fewer” should be replaced by less.

Response: Thanks for the reviewer’s suggestion, the fewer was replaced by less in Introduction of the revised manuscript.

  • Line 111, “on” makes the sentence not understandable. It also makes the following lines (112-115) hard to understand. Please reform.

Response: Thanks for the reviewer’s suggestion, the word “on” was removed, and the sentence was modified in the Introduction of revised manuscript.

  • Line 128, “normal or hydrophobic” Again, same as line 12.

Response: Thanks for the reviewer’s suggestion, the normal was replaced by neutral in the revised manuscript.

  • Line 262, Surface/area roughness parameters are Sa, Sq. etc. Line/profile roughness parameters are Ra, Rq, etc. Please note the difference. This error occurs in the manuscript multiple times! (lines 288, 336, 349, 354, 359…)

Response: Thanks for the reviewer’s suggestion, the surface roughness and the line roughness were distinguished in the Section4.2-4.4 of the revised manuscript.

  • Line 386, Roughness units missing, please add units.

Response: Thanks for the reviewer’s suggestion, the unit of roughness was added in the revised manuscript.

  • Line 387, “The CA of the droplet decreased with the increase in repetition times” By increasing the repetition times from 3 to 5 the CA increases, making this statement questionable. Reform.

Response: Thanks for the reviewer’s suggestion, the description was modified. Therein, the CA of the droplet decreased when the repetition times increased from 1 time to 3 time, which may have been caused by the droplet infiltrating into the deeper microstructures of the ablated pattern. However, the CA slightly increased when the repetition times increased from 3 time to 5 time because the smaller line roughness of 5 time ablation than 3 time ablation caused by the thermal effect. Please find the in the Section 4.4 of the revised manuscript.

  • Line 432, “normal status” Again, same as line 12.

Response: Thanks for the reviewer’s suggestion, the normal was replaced by neutral in the revised manuscript.

  • I am also missing the information about the liquid used for measuring CA? This should be added to “3.2.2. Surface morphology observation and CA measurement” section.

Response: Thanks for the reviewer’s suggestion, the deionized water was the liquid employed to measure the CA and the description was added in the Section 3.2.2 of the revised manuscript.

The revised paper has been resubmitted to you, please see the attachment. We are looking forward to your positive response.

Hsin-Yi Tsai

Associate Researcher

Taiwan Instrument Research Institute, National Applied Research Laboratories, Taiwan

Round 2

Reviewer 1 Report

The authors have improved the manuscipt following my suggestions.

I just suggest to improve the english style and  check the references.